# Dermal Drivers of Injury-Induced Inflammation: Contribution of Adipocytes and Fibroblasts

**DOI:** 10.3390/ijms22041933

**Published:** 2021-02-16

**Authors:** Paula O. Cooper, MaryEllen R. Haas, Satish kumar R. Noonepalle, Brett A. Shook

**Affiliations:** Department of Biochemistry and Molecular Medicine, School of Medicine and Health Sciences, The George Washington University, Washington, DC 20037, USA; pcooper314@gwu.edu (P.O.C.); mehaas@gwu.edu (M.R.H.); snoonepalle@gwu.edu (S.k.R.N.)

**Keywords:** inflammation, adipocyte, fibroblast, wound healing, diabetes, aging

## Abstract

Irregular inflammatory responses are a major contributor to tissue dysfunction and inefficient repair. Skin has proven to be a powerful model to study mechanisms that regulate inflammation. In particular, skin wound healing is dependent on a rapid, robust immune response and subsequent dampening of inflammatory signaling. While injury-induced inflammation has historically been attributed to keratinocytes and immune cells, a vast body of evidence supports the ability of non-immune cells to coordinate inflammation in numerous tissues and diseases. In this review, we concentrate on the active participation of tissue-resident adipocytes and fibroblasts in pro-inflammatory signaling after injury, and how altered cellular communication from these cells can contribute to irregular inflammation associated with aberrant wound healing. Furthering our understanding of how tissue-resident mesenchymal cells contribute to inflammation will likely reveal new targets that can be manipulated to regulate inflammation and repair.

## 1. Introduction

Skin is a complex, multilayered organ that provides protection from the external environment. Skin can be divided into a thin outer epidermis and thick underlying dermis. The epidermis is a stratified epithelial layer that is largely composed of tightly interconnected keratinocytes that generate a watertight barrier to prevent invasion and damage from harmful environmental agents [1]. The underlying dermis contains dozens of unique cell types and is composed of flexible extracellular matrix (ECM) components, such as collagen and elastin. The dermis can be further divided into a loose, highly vascular superficial papillary layer, an ECM-dense reticular layer, and deep dermal white adipose tissue (DWAT) [2,3]. While adipocytes and fibroblasts influence numerous aspects of skin homeostasis and support hair follicle growth [4,5,6,7], these cells also actively participate in tissue repair following injury [8,9,10,11,12,13].

Disruption of the skin’s barrier function can have catastrophic consequences by allowing harmful pathogens to invade the body. To prevent dehydration and infection, skin has evolved to rapidly respond to injury and reseal the epidermis. Upon injury, tissue-resident cells mount robust cellular and molecular responses in coordination with recruited immune cells (reviewed in [14,15]). This injury-response process has been well characterized as a sequence of overlapping phases, with each phase performing specific functions to promote efficient repair. Within minutes after injury, platelet activation and diversion of blood flow away from the site of injury minimizes blood loss. This process, known as hemostasis, is followed by an inflammatory phase that spans multiple days after injury. During inflammation, keratinocyte stores of interleukin 1α (IL1α) are released [16,17], triggering an inflammatory chain reaction by adjacent keratinocytes [18,19]. This in turn promotes an influx of neutrophils, monocytes, and macrophages to the site of injury [20,21]. Recruited neutrophils and inflammatory macrophages clear cellular debris and pathogens while perpetuating inflammation through the release of cytokines such as chemokine (C-C motif) ligand 5 (CCL5), IL1, IL6, and tumor necrosis factor α (TNFα) [15,20,22]. As inflammation actively resolves, immune cells produce vascular endothelial growth factor (VEGF) to initiate revascularization [23,24]. A subsequent phenotypic switch in macrophage polarization is regulated by many factors, including signaling from mesenchymal stem cells [25]. The transition from inflammatory to anti-inflammatory macrophage polarization supports continued tissue repair into the proliferative phase, where reparative signals orchestrate re-epithelialization and repopulation of the dermal compartment. In fact, multiple subsets of anti-inflammatory macrophages produce transforming growth factor β (TGFβ) [14,26], which is critical for activation of fibroblasts into ECM-producing myofibroblasts. The newly generated tissue, frequently a scar in adult mammals, undergoes a remodeling phase. This tissue maturation process attempts to restore the cellular and ECM composition to what existed prior to injury; however, numerous skin components, such as epidermal accessory structures (e.g., hair follicles) and deep dermal structures (e.g., DWAT), are typically not regenerated in the repaired region [9,12].

Frequently, diseases associated with impaired wound healing do not properly activate early inflammatory pathways or do not fully resolve inflammation, and therefore do not successfully progress into the proliferative phase. A delayed or incomplete transition from the inflammatory phase to the proliferative phase is associated with the persistence of inflammatory neutrophils and macrophages [27,28,29], contributing to chronic or non-healing wounds. These hard-to-treat wounds pose a significant medical challenge; as their prevalence has steadily increased over time and only modest therapeutic advancements have come from animal studies [30,31]. While tremendous efforts have uncovered defects in cellular composition and function during the proliferative phase of repair, animal models have recently revealed that reduced activation of early inflammatory responses is associated with delayed healing [32,33,34]. Due to their role in ECM production, dermal mesenchymal cells have been studied in the context of ECM formation and maturation; however, emerging evidence has revealed that adipocytes and fibroblasts can also promote inflammation. Their pro-inflammatory function is well supported in various in vivo disease models and in vitro studies that have unveiled tremendous cytokine production in response to pro-inflammatory stimuli. Below, we discuss how these abundant skin-resident mesenchymal cells play an active role in acute and chronic inflammation that follows injury.

## 2. Contribution of Adipocytes to Inflammation

### 2.1. White Adipose Tissue

White adipose tissue (WAT) is found throughout the mammalian body in various depots. While visceral (VWAT) and subcutaneous WAT (SWAT) are widely studied due to their role in metabolic disease, WAT exists in many other depots including muscle, mammary gland, bone marrow, and skin [35,36]. There are major distinctions in structure, composition, and function between individual WAT depots [9,13,37,38,39]; however, they are all predominantly composed of mature white adipocytes, immature adipocyte precursors, immune cells and blood vessels. White adipocytes maintain energy homeostasis by storing excess nutrients as triglycerides through lipogenesis and breaking down stored lipids via lipolysis during times of metabolic need. In addition to energy storage, adipose tissue has potent endocrine activity that is achieved through the release of growth factors, cytokines, and inflammatory factors often referred to as “adipokines” [40,41,42]. Adipocytes directly influence the immune cell composition and activity in and around WAT through secreted pro- or anti-inflammatory adipokines and lipids [42,43,44,45] and expression of immune checkpoint proteins [46]. For instance, human omental adipocytes constitutively express the chemokines CCL2 (monocyte chemoattractant protein 1, MCP1), and IL8/chemokine (C-X-C motif) ligand 8 (CXCL8) [47], and subcutaneous adipocytes produce adiponectin, CCL3 (MIP1α), CCL5, CXCL1, CXCL5, and leptin [48]. Notably, while macrophages and neutrophils exhibit pro-inflammatory responses when stimulated with leptin [49,50], adiponectin promotes anti-inflammatory macrophage polarization [51]. Consistent with their visceral and subcutaneous counterparts, dermal adipocytes also influence their surrounding tissues through adipokine secretions [5,52], and possess similar immune regulatory capabilities [9,13,53,54].

### 2.2. Dermal Adipocytes

DWAT has historically been considered subcutaneous tissue [3], leading to some overgeneralizations. While WAT depots have significant overlap in structure and function, key differences exist between SWAT and DWAT [9,13,39]. Many of these differences implicate dermal adipocytes as potent modulators of local immune responses [9,53]. For example, when compared to subcutaneous adipocytes, dermal adipocyte triglyceride stores are enriched with lipids capable of regulating inflammation [9] and dermal adipocytes uniquely express *Ccl4* (macrophage inflammatory protein 1 β, MIP1β), and secrete cathelicidin antimicrobial peptide (CAMP) to combat infection [13,53]. In humans, DWAT exists as a relatively thin superficial layer above SWAT [13]. Interestingly, macrophages preferentially infiltrate superficial subcutaneous WAT in humans [54], suggesting that DWAT has a greater propensity to recruit macrophages and plays a potentially prominent role in host defense.

### 2.3. WAT Inflammation

Supporting their role in immune regulation, adipocytes are equipped with receptors that sense and respond to inflammatory cues. Human and murine adipocytes express toll-like receptors (TLRs) that respond to both fatty acids and pathogen-associated molecular patterns (PAMPS) [55,56,57]. Notably, subcutaneous human adipocytes express high levels of TLR4, allowing them to respond rapidly to lipopolysaccharide (LPS) or other bacterial stimuli [55]. TLR signaling in adipocytes activates the pro-inflammatory nuclear factor kappa B (NF-κB) pathway, and stimulation with LPS results in the production of various cytokines that promote inflammation, such as CCL3, CXCL10, intercellular adhesion molecule 1 (ICAM1), IL6, IL8/CXCL8, and TNFα [55,56].

Adipocytes not only produce TNFα; they also express both receptors (TNFR1 and TNFR2) [58], and respond to TNFα in a feedforward cycle that contributes to adipose tissue dysfunction during metabolic disease [59]. Indeed, in vivo studies have linked circulating TNFα to decreased adiponectin production [60]. In vitro, TNFα treatment increased adipocyte basal lipolysis while lowering hormone-sensitive lipase (HSL) expression [61], altering glucose metabolism [58], and increasing IL1β and TLR2 expression in as little as 3 hours [57,62]. These changes in pro-inflammatory signals can be especially impactful during the early stages of wound healing.

Adipocytes also respond to IL1 ligands, as IL1β reduces insulin sensitivity in cultured human and murine adipocytes [63]. Notably, IL1 signaling can also modulate adipocyte lipolysis in vitro [64]. These data clearly demonstrate that adipocytes express receptors that integrate and propagate inflammatory signaling networks. How dermal adipocytes utilize these pathways during efficient and impaired healing is another intriguing aspect of wound healing that is actively unfolding. 

#### 2.3.1. Neutrophil Recruitment

WAT is well characterized in its ability to recruit neutrophils [65], and it is thought that these early infiltrators contribute to subsequent macrophage inflammation in adipose tissue [66]. Consistently, neutrophil infiltration is one of the first changes in adipose tissue that is caused by high-fat dieting in mice [67,68]; and in humans, increased adipose tissue abundance is correlated with increased circulating markers of neutrophil activity such as neutrophil elastase [69]. WAT can communicate with neutrophils through both direct and indirect interactions [65,67]. For example, neutrophils possess leptin receptor [50], which exerts potent pro-inflammatory activity [70] and acts as a chemoattractant [71]. Neutrophils also express free fatty acid receptors such as G protein-coupled receptor 84 (GPR84) [72], and are canonically recruited by the fatty acid-derived leukotriene b4 [73]. While crude lipid extracts from human adipocytes rapidly recruit neutrophils [74], lipolysis in VWAT also induces neutrophil recruitment and IL1β expression [65]. Specifically, oleic acid, the most abundant free fatty acid (FFA) in humans [75], recruits neutrophils to the peritoneal cavity in an IL1 receptor-dependent manner [76]. Whether similar or distinct mechanisms are utilized by dermal adipocytes during wound healing remains a topic of great interest.

#### 2.3.2. Macrophage Recruitment and Polarization

In addition to neutrophil recruitment, adipocytes directly regulate macrophage recruitment and polarization [66]. In vivo, a positive correlation exists between adipocyte size and macrophage numbers [77]. In vitro, differentiated adipocytes secrete numerous molecules that recruit macrophages including CCL3, CCL4, CCL5, and colony stimulating factor (CSF) [56]; and macrophages respond by encircling apoptotic WAT adipocytes [78]. In addition to immune-modulating adipokines, the impact of adipocyte lipid signaling is also emerging as a formidable mechanism of immune regulation [79,80]. Specifically, oleic acid can recruit macrophages and induce macrophage IL1α production [74,76], and adipocyte-derived palmitate increases macrophage TNFα production [81]. Moreover, macrophages express numerous fatty acid receptors that trigger both pro- and anti-inflammatory responses necessary for wound healing [81,82,83,84]. This suggests that dermal adipocyte-derived lipids might regulate anti-inflammatory and reparative processes in addition to early inflammatory events.

### 2.4. Adipocyte Response to Injury

DWAT is tremendously dynamic; expanding and regressing while contributing to hair follicle growth [4], cold stress [85], bacterial infection [53], and injury [8,9,13]. More recently, mammalian adipocytes have been recognized for their contributions to reduced scarring in large wounds [12]. Genetic lineage tracing experiments have revealed astounding plasticity of dermal adipocyte conversion into fibrogenic myofibroblasts after injury [9,13] and in a mouse model of fibrosis [86]. Interestingly, fat body cells, the *Drosophila* equivalent to adipocytes, actively migrate towards the site of injury to help seal wounds [87], demonstrating a conserved contribution of adipocytes to injury responses.

While systemic adipokines, such as adiponectin and leptin, promote re-epithelialization [88,89], recent efforts have been made to define the local contribution of DWAT to the injury response. Studies with fat-less A-ZIP/F-1 mice suggest that mature adipocytes are required for efficient fibroblast recruitment during the proliferative phase of repair [8]. Furthermore, blocking adipogenesis using peroxisome proliferator-activated receptor gamma (PPARγ) inhibitors GW9662 and bisphenol A diglycidyl ether (BADGE) resulted in similarly disrupted repair [8]. Consistently, adipocyte spheroid-derived secretions are sufficient to activate dermal fibroblasts into myofibroblasts [90]. To temporally regulate WAT ablation and prevent insulin resistance that occurs in constitutive mouse models [91], Zhang et al. utilized FAT-ATTAC mice, which undergo induced apoptosis of adipocytes through activation of caspase 8. Wounds in these mice healed slower, with diminished collagen deposition and delayed keratinocyte-mediated re-epithelialization [13]. These studies demonstrate that adipocytes are essential for reparative functions during the profibrotic proliferation phase. Unfortunately, manipulating adipocytes systemically makes it challenging to determine the contribution of adipocytes from specific depots. Additionally, these reports largely focus on the proliferative and remodeling phases of healing, leaving unanswered questions regarding the role of dermal adipocytes during early injury responses.

To spatially and temporally control dermal adipocyte ablation, we previously utilized a genetic mouse model of diphtheria toxin-mediated adipocyte cell death [9]. We discovered that dermal adipocytes were required to support efficient revascularization and epithelial repair during the proliferation phase of repair, and that ablation of dermal adipocytes resulted in a 50% reduction in inflammatory wound bed macrophages 1.5-days after injury [9]. Further examination revealed that the DWAT undergoes hypertrophic expansion shortly after injury [9], similar to what is observed following *Staphylococcus aureus* infection [53]. After this initial expansion, wound bed adipocytes undergo lipolysis and revert to their original size concomitant with macrophage infiltration. Quantitative lipidomic analysis revealed palmitoleic acid, oleic acid, α-linoleic acid and medium-chain fatty acids as major products of injury-induced dermal adipocyte lipolysis [9]. Interestingly, these fatty acids have been implicated in regulating macrophage inflammation [74,76,92]; and when dermal adipocyte lipolysis was impaired in mice lacking adipose triglyceride lipase (ATGL), fewer inflammatory macrophages were detected [9] (Figure 1). Though the mechanism by which lipolysis-mediated signaling supports the inflammatory macrophage response after injury remains elusive, it is clear that dermal adipocyte-derived lipids are capable of regulating this response.

## 3. Contribution of Fibroblasts to Injury-Induced Inflammation

### 3.1. Contribution of Fibroblasts to Tissue Inflammation

Since activated wound bed myofibroblasts are the main producers of ECM [93], they have been extensively studied during the proliferative and remodeling phases of tissue repair. Recent discoveries have demonstrated that fibroblasts also play an active role in tissue inflammation. Following injury, fibroblasts contribute to early inflammatory pathogen and damage responses in numerous tissues, such as skin, lung, liver, intestines, heart, conjunctiva, urogenital tract and adipose tissue [94,95,96,97,98]. These pro-inflammatory fibroblasts contribute to the immune response, frequently through the recruitment and activation of myeloid cells. After inflammation subsides, fibroblasts mediate ECM deposition, indicating that fibroblasts can exist in a pro-inflammatory, profibrotic axis, similar to macrophages and keratinocytes. While direct in vivo exploration of interactions between dermal fibroblasts and immune cells is in its infancy, the inflammatory nature of fibroblasts has been clearly demonstrated in other tissues.

Multiomic characterization of murine fibroblasts from multiple organs recently illuminated an underappreciated immune function of these structural cells [94]. Transcriptional analysis of dermal fibroblasts revealed enrichment for ligands and receptors that predict a propensity for B cell, macrophage, and monocyte interactions. Subsequent Assay for Transposase-Accessible Chromatin (ATAC) sequencing demonstrated transcription potential at multiple immune gene loci in dermal fibroblasts, including interferon gamma receptor 1 *(Ifn*γ*r1*) [94]. Furthermore, chromatin accessibility and gene enrichment cross-referencing predicted that dermal fibroblasts are poised to rapidly transcribe genes associated with antigen processing and presentation, complement and coagulation cascades, and sphingosine-1-phosphate signaling pathways. In another study, single-cell RNA sequencing (scRNA-seq) was performed on fibroblasts from healthy human skin and samples from inflammatory diseases (acne, alopecia areata, granuloma annulare, leprosy, and psoriasis) [95]. Fibroblasts formed nine transcriptionally-distinct clusters with fibroblast composition varying greatly across disease types; however, many immune genes were upregulated in multiple clusters such as *CCL2, CCL19, CXCL12, CXCL14, IL6,* and *IL8/CXCL8* [95]. These studies highlight the broad pro-inflammatory capacity of dermal fibroblasts. Interestingly, proteomic analysis of fibroblasts from psoriatic patients confirms higher levels of inflammation-associated proteins, such as TNFα [99] and supernatant from psoriatic fibroblasts induces an inflammatory macrophage phenotype [100]. Additionally, fibroblasts from atopic dermatitis patients induce inflammatory gene expression in cultured skin equivalents [101]. Since cultured human dermal fibroblasts upregulate *CCL2, CCL7,* and *IL6* when stimulated with supernatant from inflammatory macrophages [102], it is likely that injury-associated signaling activates a pro-inflammatory phenotype in dermal fibroblasts.

Additional insights can be gained from the injury response in cardiac fibroblasts. Following myocardial infarction (MI), cardiac tissue progresses through an inflammation-to-repair transition similar to skin repair. Gene expression analysis of cardiac fibroblasts 1 day after MI revealed upregulation of inflammatory cytokines, such as *Ccl5,* and *Cxcl3*, coupled with a downregulation of TGFβ signaling component genes [98]. Furthermore, primary cultured cardiac fibroblasts from severe heart failure patients exhibited LPS-induced cytokine production with increased expression of CCL2, IFNγ, IL1β, IL6, IL8/CXCL8, and TNFα [103]. By 3 days after MI, cardiac fibroblasts transition towards proliferative and pro-angiogenic function before shifting toward a collagen and fibronectin depositing, profibrotic function 7 days after injury [104]. Due to the strong transcriptional similarity between dermal fibroblasts and cardiac fibroblasts [94,105], it is likely that dermal fibroblasts contribute to inflammation through mechanisms parallel to cardiac fibroblasts.

### 3.2. Signaling Pathways Regulating Inflammatory Fibroblast Phenotype

While the direct influence of fibroblasts to injury-induced inflammation has been limited by the genetic tools available, in vitro studies have revealed remarkable potential for fibroblasts to produce pro-inflammatory signaling molecules. Fibroblasts upregulate pro-inflammatory gene expression following stimulation from numerous cytokines present during skin wound healing, such as IFNγ, IL1α, IL1β, and TNFα [22,106,107] (Figure 1); and in response to irritants capable of inducing skin inflammation, such as PM10 and cobalt chloride [108,109]. The results from these studies provide valuable insights into the pro-inflammatory capacity of fibroblasts.

#### 3.2.1. IL1 Signaling

The alarmin cytokine IL1α promotes an inflammatory fibroblast phenotype in cultured primary fibroblasts from human lung [110] and infrapatellar fat pad [111]. IL1α stimulation leads to NF-κB pathway activation in fibroblasts with subsequent production of CCL2, IL6, and IL8/CXCL8 [110,111]. Through a series of Transwell migration assays, Paish et al. (2018) showed that IL1α-stimulated fibroblasts promote monocyte recruitment through CCL2 [111]. Importantly, keratinocyte-derived IL1α induces dermal fibroblast secretion of CCL2, CXCL1, IL6, and IL8/CXCL8 [112], and co-stimulation with IL1α and TNFα synergistically enhances the pro-inflammatory phenotype of dermal fibroblasts [112]. Since both IL1α and TNFα are rapidly-released by keratinocytes during the early injury response [17], this interaction could instigate fibroblast cytokine production following injury, though interrogations of fibroblast responses to IL1α signaling in vivo are necessary to functionally validate these findings in an injury-based context.

IL1β modulates inflammation by both inducing pro-inflammatory gene transcription and extending mRNA longevity and translation potential through mRNA transcript stabilization [113,114]. In both human and canine cultured dermal fibroblasts, IL1β stimulation induces transcription of *IL8/CXCL8* and *IL6* in a dose-dependent manner and elevates IL6 at the protein level [114,115]. Fibroblast *IL6* upregulation is further amplified by IL1β/TNFα co-stimulation [114], which is important to note due to the myriad of pro-inflammatory mediators simultaneously released during the early injury response. This pro-inflammatory response to IL1β is observed by fibroblasts derived from numerous tissues, including embryonic lung [114], shoulder capsule [116], and peritoneal tissue [113]. Indeed, IL1β activates non-dermal fibroblasts to produce CCL20, IL6, and IL8/CXCL8 [116] as well as polymorphonuclear leukocyte (PMN)-attractant chemokines CXCL1, granulocyte colony stimulating factor (GCSF), and IL8/CXCL8 [113]. These results demonstrate a well-conserved pro-inflammatory response to IL1 signaling in fibroblasts.

#### 3.2.2. TNFα Signaling

Shortly after injury, wounded skin is enriched with high levels of TNFα derived from multiple cell types [117]. In vitro, human dermal fibroblasts respond to acute TNFα stimulation through rapid expression of potent pro-inflammatory and myeloid cell recruitment factors, such as CCL2, CCL3, CCL4, CXCL1, *CXCL8*, CXCL12, *Il1β*, IL6, serine proteinase inhibitor 1 (*SERPINE1*)*,* and *TNFα* [106,118]. While inducing a pro-inflammatory fibroblast polarization, TNFα suppresses fibroblast proliferation, migration, and transition toward the profibrotic myofibroblast phenotype [119], and induces the expression of matrix metalloproteinase 9 (*MMP9*) in dermal fibroblasts and active MMP2 in collagen-latticed fibroblast cultures [106,120].

Fibroblasts also express cluster of differentiation 40 (CD40), a TNFα receptor family member capable of transducing pro-inflammatory signals. The CD40 receptor is expressed by human fibroblasts isolated from lung, spleen, skin, synovium, gingiva, and periodontal ligament [121,122,123]. CD40 activation initiates the NF-κB pathway, culminating in the transcriptional upregulation of pro-inflammatory genes such as *CCL2, CCL5, ICAM1, IL6,* and *IL8/CXCL8* [122,124,125]. Inflamed gingival tissue contains greater amounts of CD40 relative to controls, and fibroblasts upregulate CD40 expression in response to IFNγ [122], including a 10-fold CD40 transcriptional increase in cultured dermal fibroblasts 24 hours after stimulation [121]. In cultured lung fibroblasts, CD40-CD40L interactions result in the upregulation of cyclooxygenase-2 (COX2) and prostaglandin E2 (PGE2) [123]. When stimulated by both IFNγ and CD40L, fibroblasts dramatically increased COX2 and PGE2 over 12-fold [123]. COX2 and PGE2 are associated with tissue inflammation, with PGE2 acting as a chronic inflammatory mediator and fibrotic inhibitor [93,123]. Interestingly, in the absence of a disease phenotype, dermal fibroblast-derived COX2 and PGE2 support a transition from inflammatory to anti-inflammatory macrophage polarization in vitro [126]. These findings suggest that IFNγ stimulates fibroblasts to increase expression of CD40, which responds to CD40L on infiltrating activated immune cells to support a pro-inflammatory secretory profile [123].

#### 3.2.3. TLR Signaling

Human dermal fibroblasts constitutively express TLRs 1–10 [127,128]. Stimulation of dermal fibroblasts with TLR1/2, 3, and 4 ligands activates extracellular signal-related kinase (ERK) and NF-κB cascades [128], resulting in the production of IL6 and IL8/CXCL8 [128]. Interestingly, compared to keratinocytes, dermal fibroblasts were shown to be transcriptionally-enriched for TLRs [128], with greater protein enrichment for TLR2, TLR3, and TLR4 [128]. A strong damage/pathogen response role of skin fibroblasts was observed in vitro through the targeted stimulation of TLR-1 and 2 with a synthetic triacyl lipopeptide (Pam3CSK4) [128]. Both 10 and 24 hours after stimulation, fibroblasts were enriched for *IL6* and *IL8* mRNA compared to keratinocytes [128]. Similarly, cultured human dermal fibroblasts showed elevated CCL2, IL8/CXCL8, and NF-κB signaling in response to a 72-hour stimulation with LPS, the major ligand for TLR4 [109]. Since dermal fibroblasts are capable of mounting a powerful response to various damage or pathogen-derived molecular cues, future lines of investigation will likely reveal that TLR signaling activates a pro-inflammatory fibroblast profile in additional inflammatory contexts.

### 3.3. Molecular Regulation of Fibroblast Polarization

Since fibroblasts can have a pro-inflammatory or profibrotic phenotype, researchers have begun investigating molecular modulators that regulate fibroblast polarization. The PU.1 transcription factor is associated with a profibrotic phenotype in fibroblasts from various tissues, including skin, lung, liver and kidney [129]. In inflammatory fibroblasts PU.1 is post-transcriptionally inhibited through micro-RNA 155 (miR-155), which induces a pro-inflammatory profile in fibroblasts and inhibits fibroblast proliferation [129,130]. The pro-inflammatory effect of miR-155 was observed to enhance CCL2, IL1β, IL6, and TNFα expression in cardiac fibroblasts in vitro and following MI [130]. Consistent with a pro-inflammatory, profibrotic polarization axis, miR-155-null mice exhibited greater collagen deposition and numbers of fibroblasts enriched for *α* smooth muscle actin (*α*SMA/actin alpha 2, *Acta2*), collagen 1 (Col1/Col1*α*), and *Col3α* [130]. These diverging fibroblast profiles illustrate a bi-faceted fibroblast role in inflammation and fibrosis that is modulated by miRNA-155 and PU.1. In addition to PU.1, the transcription factor early B-cell factor 2 (EBF2) can influence the gene expression profile of fibroblasts, as *Ebf2* depletion results in decreased expression of fibroblast activation genes *Acta2*, *Il6* and *Ccl1* [131].

### 3.4. Functional Diversity in Fibroblasts

Lineage tracing, RNA sequencing and cellular surface marker profiling have defined tremendous dermal fibroblast heterogeneity in uninjured skin and wound beds [10,11,132]. This vast heterogeneity between fibroblast subsets is coupled with functional diversity during repair. In particular, fibroblasts residing in the papillary dermis can regenerate arrector pili muscles and dermal papillae [132,133]; and deep dermal and subcutaneous fascia fibroblasts expressing dipeptidyl peptidase 4 (*Dpp4*)/CD26 can differentiate into ECM producing myofibroblasts [10,11,132,133,134].

Though in vivo data examining functional cellular heterogeneity during the inflammatory phase of wound healing are lacking, we previously reported differential expression of pro-inflammatory genes in fibroblast subsets 5 days after injury. Specifically, elevated *Ccl2* expression was observed in stem cell antigen 1 (SCA1)^+^; CD34^+^; CD26^+^ fibroblasts and elevated *Il1α*, *Il1*β, and *Tnf* expression was observed in Sca1^−^; CD34^−^; CD29^high^ fibroblasts [10]. Similarly, in murine and human skin, a pro-inflammatory gene expression profile was observed within fibroblasts residing in the reticular dermis, which illustrates a possible inflammatory inclination of specific fibroblast populations [135]. Not only do these findings implicate a pro-inflammatory role for fibroblasts as early responders and modulators of the injury response, but they also highlight the possibility that cellular diversity contributes to differential inflammatory responses during wound healing.

Similar to skin, scRNA-seq of VWAT revealed heterogeneity among mesenchymal cells, with a distinct lymphocyte antigen 6 complex (LY6C)^+^; platelet-derived growth factor β (PDGFRβ)^+^ population enriched for both inflammatory and fibrotic genes [136]. In vitro, these “fibro-inflammatory” progenitors produced *Col1* and *Col3* in response to TGFβ signaling. Similarly, both LPS and TNFα stimulation promoted increased gene expression of inflammatory signaling molecules *Ccl2*, *Cxcl2*, *Cxcl10*, and *Il6* [136]. Pro-inflammatory fibroblasts were able to activate macrophage inflammatory gene expression following in vitro stimulation and in vivo during high-fat dieting, contributing to immune cell recruitment [136]. Interestingly, these responses were significantly greater than the response generated from tissue-resident adipocyte precursor cells. Similar functional diversity has been observed using scRNA-seq in rheumatoid arthritis and osteoarthritis. Podoplanin (PDPN)^+^; CD34^+^; thy-1 cell surface antigen 1 (THY1)^+^ synovial fibroblasts are enriched for pro-inflammatory gene expression, and robustly producedCCL2, CXCL12, and IL6 when stimulated with TNFα in vitro [137]. In another report, PDPN^+^; fibroblast activation protein α (FAPα)^+^; THY1^+^ fibroblasts promoted persistent and severe joint inflammation, immune cell recruitment, and production of IL6, IL33, IL34, and leukemia inhibitory factor (LIF) [138]. These data support that specific fibroblast subsets may be biased in their ability to elicit inflammatory responses. While further investigation is required to define the role of individual fibroblast populations to injury-induced inflammation, it is likely that biases in the pro-inflammatory, profibrotic capacity of fibroblast subsets contribute to contrasting phases of inflammation.

### 3.5. Communication between Adipocytes and Fibroblasts

In addition to direct interactions with immune cells, there is substantial crosstalk between dermal fibroblasts and adipocytes. Indeed, human dermal fibroblasts express receptors for numerous adipokines, including leptin and adiponectin [139]. Consistent with its anti-inflammatory properties, adiponectin plays an attenuative role in dermal fibrosis through reducing fibroblast activation [140]. Furthermore, UV exposure associated with aging decreases dermal adipocyte production of leptin and adiponectin, which in turn reduces dermal fibroblast production of pro-inflammatory TNFα [141]. Contrastingly, UV irradiated fibroblast conditioned media increased dermal adipocyte expression of pro-inflammatory cytokines including CCL5, CCL20, and CXCL5 in vitro [48]. These findings suggest that communication between adipocytes and fibroblasts likely contributes to their pro-inflammatory function after injury.

## 4. Altered Inflammatory Response during Impaired Wound Healing

Aging and diabetes are associated with a myriad of skin conditions, the most predominant of which is delayed wound healing [142,143]. Elderly and diabetic individuals are susceptible to chronic wounds, with up to 25% of type 2 diabetics experiencing difficulties with healing [142,144]. Both aged and diabetic skin feature alterations in ECM, including irregular collagen cross-linking [145,146] and increased disintegration associated with greater MMP activity [146,147,148] that contribute to impaired wound healing [142,149]. While this diminished fibrotic capacity could reduce scar formation [11,150], it often leads to chronic inflammation by allowing bacterial [151,152] or fungal [153] overgrowth with a subsequent overproduction of cytokines and proteases [154,155]. Since chronic wounds can persist for over a year and are frequently observed in an inflammatory state [155], studies have historically focused on factors that promote reparative processes during the proliferative phase in control groups. These studies produced prospective targets for improved healing outcomes, including administration of mesenchymal stem cells to dampen inflammation and promote ECM production [156]. Interestingly, new lines of investigation have uncovered a need for robust, efficient recruitment of leukocytes to support proper repair [33,34,157], making factors that impact early inflammation a critical area of research. In particular, delayed or sustained neutrophil or macrophage function can have detrimental effects on multiple facets of downstream wound resolution and healing [158,159].

### 4.1. Impaired Early Leukocyte Infiltration and Function

Although early healing time points can be challenging to obtain from humans, diabetic mouse studies have detected epigenetic- [160] and chemokine-mediated [157] delays in macrophage recruitment and activation at early time points after injury. Thorough analysis of wound bed myeloid cells revealed a marked delay in peak macrophage numbers of diabetic mice as well as various changes in transitioning immune cells [33,34]. In the elderly population, lower basal hematopoiesis [161] may compound decreased macrophage responsiveness and inflammatory polarization [162,163]. Notably, delayed macrophage infiltration was observed in human wound biopsies from aged individuals [164] and macrophages in wounds of aged mice lack proper phagocytic activity [165].

### 4.2. Persistence of Inflammation

In addition to a delay in the initial macrophage response, a second influx of inflammatory macrophages impairs healing in high-fat diet-induced diabetic mice [166]. Both diabetes and aging are characterized by systemic inflammation [167,168], likely contributing to persistence of inflammatory neutrophils and macrophages at later time points after injury [28,33]. Pro-inflammatory skewing of diabetic macrophages [169,170] begins in the bone marrow [171], and macrophages from aged mice have a diminished capacity to respond to external polarization signals [162], reducing their ability to transition during repair. Consistently, elevated levels of pro-inflammatory macrophage chemoattractants have been identified in human chronic wounds [172,173]. The resulting inflammation is exacerbated due to reduced numbers and function of anti-inflammatory cells, such as mesenchymal stem cells [156]. This pro-inflammatory environment prevents macrophages from transitioning into anti-inflammatory macrophages, leading to recalcitrant inflammation [27] that prevents proper transition into the proliferative and remodeling phases of repair.

## 5. Contribution of Adipocytes to Impaired Wound Healing

### 5.1. Diabetes-Associated Changes in Adipocyte Inflammatory Function

An estimated 83% of diabetic individuals are overweight or obese, with the highest prevalence of diabetes in the most obese individuals [174]. Diabetes- and obesity-related changes go hand in hand, as insulin resistance develops in a spectrum of systemic alterations as adipocytes increase in size and undergo functional changes related to lipid metabolism [77,175] and inflammation [176]. Although adipocyte-mediated inflammation is necessary for proper glucose metabolism and WAT expansion [177,178], it also contributes to systemic inflammation and macrophage infiltration that cause metabolic dysfunction [176,179]. While changes in VWAT have functionally been implicated in systemic inflammation associated with diabetes [180,181], WAT can also contribute to local tissue inflammation. For example, periprostatic adipocyte size is correlated with higher prostatic inflammation [182], and greater intramuscular adipose tissue is associated with increased IL6 levels and muscle inflammation [183]. As a consequence, changes in the pro-inflammatory function of dermal adipocytes likely play a role in altered inflammation during diabetic wound healing (Figure 2).

#### 5.1.1. Impaired Early Leukocyte Infiltration and Function

Larger adipocytes are less responsive to external stimuli [184,185]. Consequently, diabetes is associated with impaired stimulated lipolysis as a result of reduced expression of lipases involved in lipid catabolism [186,187]. Since obesity leads to increased dermal adipocyte size [13,85], DWAT function is likely altered with diabetes. Given that injury-induced lipolysis generates pro-inflammatory factors at the site of injury [9], impaired stimulated lipolysis can significantly reduce macrophage recruitment and the downstream phases of wound healing. In addition to reduced macrophage numbers during early stages of repair, diabetic wounds also exhibit deficiencies in macrophage polarization and function [188,189]. The emerging role of CAMP as a myeloid regulator [190] suggests that a lack of CAMP would significantly impact macrophage inflammation. Indeed, CAMP promotes phagocytosis [191] and inflammatory macrophage polarization [192]. Notably, while CAMP levels have been positively correlated with adipocyte size [193], wound from diet-induced obese mice and human diabetic foot ulcers have reduced levels of cathelicidin [194,195]. Thus, an inability of adipocytes to respond to wound-induced stimuli may decrease the pro-inflammatory response in early wound healing and impact later stages of repair.

#### 5.1.2. Persistent Inflammation

Despite decreased stimulated lipolysis, diabetics exhibit elevated basal lipolysis in visceral adipocytes, which contributes to VWAT inflammation [184,196,197,198]. Increased elevated basal lipolysis likely results in a greater concentration of pro-inflammatory fatty acids. While the initial burst of injury-induced lipolysis is necessary for macrophage inflammation [9], prolonged, elevated basal lipolysis may contribute to persistent pro-inflammatory macrophages or reduced anti-inflammatory macrophage differentiation necessary for wound resolution.

Adipokines also recruit immune cells into diabetic WAT, including neutrophils and inflammatory macrophages. These immune cells respond and contribute to increased circulating inflammatory adipokine levels [169,199], providing clues to how dermal adipocytes function may contribute to diabetic wound healing. For example, VWAT from diabetic individuals produces higher levels of CCLs that recruit macrophages [200] and pro-inflammatory factors including CCL2, IL1, IL6, IL18, Leptin, and TNFα [169,199], with lower levels of anti-inflammatory adipokines such as adiponectin and its paralogs (C1q/TNF-receptor proteins (CTRPs)) [201,202]. Similarly, as obesity increases, subcutaneous adipocytes secrete greater amounts of IL1β and TNFα [180]. These elevated basal pro-inflammatory signals may in turn prevent anti-inflammatory macrophage polarization and maintain greater neutrophil and inflammatory macrophage numbers in chronic diabetic wounds [27].

Biofilms also contribute to significant tissue destruction and sustained inflammation in diabetic wounds [203]. In addition to its potential role in early inflammation, reduced cathelicidin LL37 in diabetic wounds [194] may also contribute to biofilm control [204]. Thus, loss of adipocyte cathelicidin LL37/CAMP may promote biofilm-mediated inflammation and contribute to chronic wounds. Whether dermal adipocytes contribute directly to biofilm formation and other aspects of altered diabetic wound healing has yet to be revealed; however, their potential to alter the local inflammatory environment makes them an intriguing focus for future studies.

### 5.2. Age-Associated Changes in Adipocyte Inflammatory Function

With age, adipose tissue undergoes significant redistribution, resulting in decreased peripheral WAT and increased VWAT [205]. Additionally, aging is associated with higher baseline inflammation [168]. One major distinction between diabetes and aging is dermal adipocyte prominence. There is tremendous variability in the proportions of WAT depots throughout aging, including reported discrepancies in age-related changes in DWAT abundance in mice (discussed in [206]). Nevertheless, when gender, hair cycle, and location are accounted for, aged murine DWAT decreases in prominence [207,208] and differentiation potential [209]. In general, human DWAT also decreases in prominence with progressive aging [205,210] and elderly individuals undergo alterations in circulating adipokines [211,212]. These and other age-related changes in dermal adipocytes may alter immune function and likely contribute to defective inflammation that occurs during wound healing in the elderly (Figure 2).

#### 5.2.1. Impaired Early Leukocyte Infiltration and Function

Given the age-related decrease in DWAT size, wound healing is likely impacted by deficiencies in adipocyte-derived factors. For example, an age-related decrease in adipocyte CAMP production [209] can reduce macrophage phagocytosis [191,213] and inflammatory macrophage polarization [192], reducing the initial response to injury. Indeed, aged adipocyte precursors display impaired potential for differentiation [214,215], which is essential for CAMP production [53,209]. Additionally, aging is associated with reduced lipid storage and processing in adipocytes [216]. The combination of decreased wound-induced lipolysis and diminished DWAT prominence can result in a deficit of FFA signaling [9], compounding the impaired macrophage response in elderly individuals.

#### 5.2.2. Persistent Inflammation

Age-related changes in dermal adipocytes are likely to contribute to the persistence of inflammatory immune cells at later time points after injury. By decreasing the initial macrophage response and phagocytic ability, while simultaneously decreasing antimicrobial CAMP, bacterial infection can persist in aged skin [204,209]. This creates a condition with greater pathogen burden, requiring the persistence of pro-inflammatory macrophages and neutrophils that establish a cycle of inflammation. Additionally, in vitro, aged adipocytes have greater production of CCL2 and IL6 while simultaneously decreasing adiponectin [217]. This baseline increase in adipocyte-produced pro-inflammatory factors may directly contribute to persistent immune cell infiltration during wound healing.

## 6. Contribution of Fibroblasts to Impaired Wound Healing

### 6.1. Diabetes-Associated Changes in Fibroblast Inflammatory Function

Diabetic fibroblasts exhibit decreased proliferation, adhesion, and migration as well as altered MMP expression and ECM production [218,219]. This fibroblast dysregulation hinders fibrosis and results in significant delays in dermal repair [29,32,220]. Mounting evidence supports that impaired responsiveness and an exaggerated pro-inflammatory phenotype in fibroblasts contributes to reduced macrophage recruitment shortly after injury and subsequent persisting inflammation at later time points (Figure 2). Exciting new lines of investigation have been defining how these changes in fibroblast-driven inflammation contribute to impaired diabetic wound healing.

#### 6.1.1. Impaired Early Leukocyte Infiltration and Function

In vitro evidence suggests that diabetic human dermal fibroblasts lack responsiveness to TNFα stimulation [106]. Diminished TNFα-induced expression of *CCL2, IL6, IL8/CXCL8,* and *SERPINE1* in diabetic fibroblasts [106,157] implicates fibroblast dysregulation as a contributor to delayed macrophage recruitment during the early injury response. Diabetic fibroblasts may similarly impair neutrophil recruitment and activation, due to diminished *CXCL1* and *IL8/CXCL8* expression following stimulation [106,221]. While there is a clear gap in our knowledge of how fibroblasts respond and contribute to early inflammatory events in diabetic wounds, these findings demonstrate an impaired responsiveness by dermal fibroblasts under the diabetic diseased conditions.

#### 6.1.2. Persistent Inflammation

Diabetic fibroblasts undergo early senescence [218], and acquire a senescence-associated secretory phenotype (SASP) [222]. These changes result from genotoxic stimuli and oxidative stress [223] due to long-term TNFα [224] and glucose exposure [225]. The diabetic SASP in fibroblasts is associated with elevated expression of pro-inflammatory cytokines, chemokines, and ECM-degrading molecules [106,226]. Indeed, hyperglycemia can elevate reactive oxygen species (ROS) in fibroblasts, while increasing expression of apoptotic and pro-inflammatory genes, including (caspase 3/*CASP3*), *CCL13, IL8/CXCL8*, *SERPINE1*, and TNFα [225,227]. Furthermore, fibroblasts cultured in hyperglycemic and hypoxic conditions exhibit an innate immunity hypervigilance, namely elevated expression of TLR4, NF-κB signaling components, caspase 3, and IL6 [228]. Broadly, TLR4 hyperexpression in diabetic wounds sustains inflammation and impairs wound closure [228,229]. Recently, these findings were confirmed in primary human fibroblasts grown in hyperglycemic and hypoxic conditions, wherein blocking TLR4 improved fibroblast migration and lowered caspase 3, IL6, and high mobility group box 1 (HMGB1) secretion [228].

In addition to in vitro studies, analysis of chronic wounds demonstrates that fibroblasts can support persistent inflammation. Specifically, fibroblasts within chronic diabetic ulcers exhibit diminished heterogeneity that favors populations that are enriched for inflammation-related gene expression [29,32]. The elevated pro-inflammatory state can also be observed across different populations of fibroblasts, as diabetic ulcer papillary fibroblasts have elevated *IL11, IL24, MMP1,* and *MMP3* expression and another identified fibroblast cluster is enriched with *IL6*, *MMP12,* and prostaglandin endoperoxide synthase 2 (*PTGS2*) expression [32]. Locked in this pro-inflammatory state, diabetic fibroblasts are anti-angiogenic and antifibrotic with reduced transcription of growth factors and genes involved in proliferation and collagen organization [29,32]. This anti-angiogenic and antifibrotic polarization is epigenetically-encoded and maintained by diabetic fibroblasts after repeated passages in culture [230]. Thus, diabetic fibroblasts have impaired fibrogenic function and become affixed in a pro-inflammatory state, potentially driving persistent inflammation while resisting a profibrotic transition during wound healing.

### 6.2. Age-Associated Changes in Fibroblast Inflammatory Function

Studies of dermal fibroblasts during aging have discovered numerous changes that contribute to impaired wound healing. Elderly human skin contains fewer fibroblasts, and dermal fibroblasts exhibit reduced motility and proliferation, with substantial changes in collagen deposition [148,219]. With age, human dermal fibroblasts lose differential expression of cellular identity genes [231] and exhibit diminished fibrogenic potential through the downregulation of ECM-related genes [232]. An age-related decrease in fibroblast traction and spreading simultaneously induces a pro-inflammatory and antifibrotic effect, in which increased production of PGE2 dampens protocollagen production necessary for ECM maintenance [233]. Finally, RNA-seq analysis of fibroblasts predicts an age-related reduction in receptor-ligand interactions with other skin cell types [231], which are critical for efficient repair.

#### 6.2.1. Impaired Early Leukocyte Infiltration and Function

The age-dependent contribution of fibroblasts to impaired early inflammation is beginning to be revealed through signaling interactions with immune cells. Wall et al., assessed how cultured fibroblasts isolated from chronic wounds and normal patient-matched skin respond to a wound-mimicking stimulation [234]. Interestingly, chronic wound fibroblasts from aged individual exhibit diminished transcriptional induction of pro-inflammatory genes after in vitro wound simulation, including lower levels of *CXCL1*, *CXCL2*, *CXCL3*, *CXCL5*, *CXCL6*, *ICAM1*, and *IL1R1* [234]. Subsequent protein analysis confirmed decreased CXCL1 and CXCL5 secretion from chronic wound fibroblasts [234]. Functionally, this altered chemoattractant profile of aged chronic wound fibroblasts corresponded to delayed neutrophil recruitment within a chemotaxis assay [234]. These findings suggest that age-related changes in dermal fibroblast responsiveness contribute to delayed myeloid cell recruitment immediately after injury (Figure 2). However, heightened inflammatory responsiveness to LPS stimulation has been observed in primary dermal fibroblasts isolated from aged individuals [235]. Since age-related human studies have relied on in vitro stimulation of fibroblasts, future lines of investigation are needed to determine whether human dermal fibroblasts exhibit delayed activation in vivo after injury.

#### 6.2.2. Persistent Inflammation

Similar to what is observed with diabetes, dermal fibroblasts undergo numerous age-related changes that can support sustained inflammation (Figure 2). Dermal fibroblasts experience age-dependent telomere shortening and ROS accumulation [223], resulting in a greater number of senescent fibroblasts [147,231] and the development of a SASP [236]. Correspondingly, dermal fibroblasts feature age-related upregulation of genes associated with pro-inflammatory cytokine synthesis, leukocyte recruitment, and MMPs [147]. Notably, conditioned medium from aged murine fibroblasts shows significantly higher levels of pro-inflammatory cytokines IFNγ, IL1α, IL1β, IL2, IL6, IL18, LIF, and TNF, than young counterparts [131]. It is likely that the elevated pro-inflammatory state of dermal fibroblasts directly perpetuates inflammatory signals, resulting in persistence of neutrophils and inflammatory macrophages during wound healing. Additionally, fibroblast composition during the proliferative phase shows that aging skews wound bed fibroblasts away from profibrotic gene expression and toward pro-inflammatory cytokine production [10,131]. Studies of wound healing in aged mice revealed changes in wound bed fibroblast proliferation and heterogeneity that result in increased numbers of pro-inflammatory fibroblasts with fewer fibrogenic fibroblasts [10,131]. Specifically, wound beds from aged mice possess diminished populations of *Acta2*, *Cxcl5*, *Dpp4*/CD26, and microfibrillar associated protein 5 (*MFAP5*) expressing fibroblasts [10,131,147]. These data indicate that fibroblasts exhibit a failed pro-inflammatory to profibrotic transition with age that contributes to the delayed progression of repair.

## 7. Methods

PubMed searches were performed for different combinations of the terms “fibroblast”, “adipocyte”, “inflammation”, and “wound healing” for the period January 1900–January 2021. This resulted in greater than 39,000 total results. Manuscripts were narrowed for relevance based on providing empirical evidence that described mechanisms for how fibroblasts or adipocytes respond and contribute to inflammation. Skin studies and more recent reports received greater emphasis per the guidelines of the journal. Approximately 500 articles were found to be relevant to the topic and further examined for inclusion in the article. This review should be considered a narrative rather than a systemic review.

## 8. Conclusions and Future Directions

The ability of an organism to rapidly promote and resolve inflammation is critical to combat pathogens and promote repair. Recently, the stroma has emerged as a key component in the inflammatory response of various tissues. Growing evidence has revealed that skin-resident adipocytes and fibroblasts are two prominent dermal mesenchymal cell populations that contribute to cutaneous inflammation. Additionally, both adipocyte and fibroblast functions are altered by diseases such as diabetes and aging, in which these cells exhibit a higher transcriptional baseline of pro-inflammatory gene expression but their ability to rapidly respond to stimulatory cues is significantly dampened. Future investigations are needed to reveal the magnitude and precise molecular mechanisms connecting mesenchymal cells to inflammation in both efficient and dysfunctional inflammation. These studies will allow new lines of translational research to exploit inflammatory signaling pathways and fine-tune tissue inflammation, similar to approaches that target later stages of repair [12,93]. For example, increasing adipocyte and fibroblast responsiveness and production of cytokines that initially recruit and activate immune cells may encourage a robust influx of myeloid cells in the early phases of wound healing (Table 1). Contrastingly, by reducing adipocyte and fibroblast cytokine production during the later stages of chronic wounds, the recalcitrant pro-inflammatory cycle may be disrupted to allow for proper resolution (Table 1). Further discovery of critical signaling pathways and target cells could allow therapeutics to bypass the defects in adipocyte and fibroblast function by directly targeting the cells and receptors that they affect. This progression in our understanding of how tissue inflammation is regulated can more broadly advance the treatment of diseases associated with irregular tissue inflammation, such as cancer, metabolic disease, and infection.

## Figures and Tables

**Figure 1 ijms-22-01933-f001:**
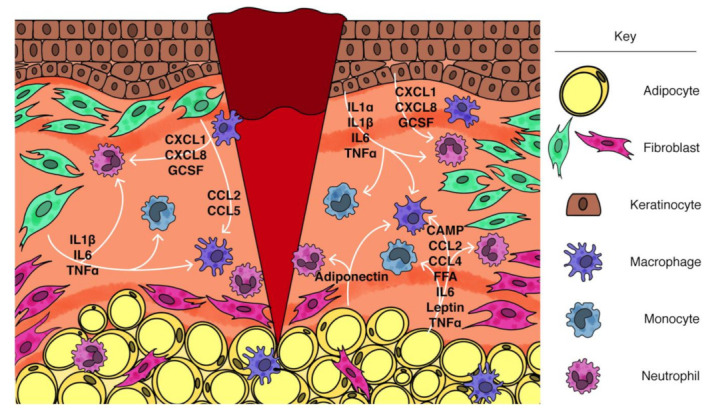
Regulation of injury-induced inflammation by skin-resident cells. After injury, skin-resident cells release factors that promote inflammation. Arrows indicate factors secreted from keratinocytes, adipocytes, and fibroblasts and the prospective leukocyte interactions during wound healing. CAMP, cathelicidin antimicrobial peptide; CCL, chemokine (C-C motif) ligand; CXCL, chemokine (C-X-C motif) ligand; FFA, free fatty acid; GCSF, granulocyte colony stimulating factor; IL, interleukin; TNF, tumor necrosis factor.

**Figure 2 ijms-22-01933-f002:**
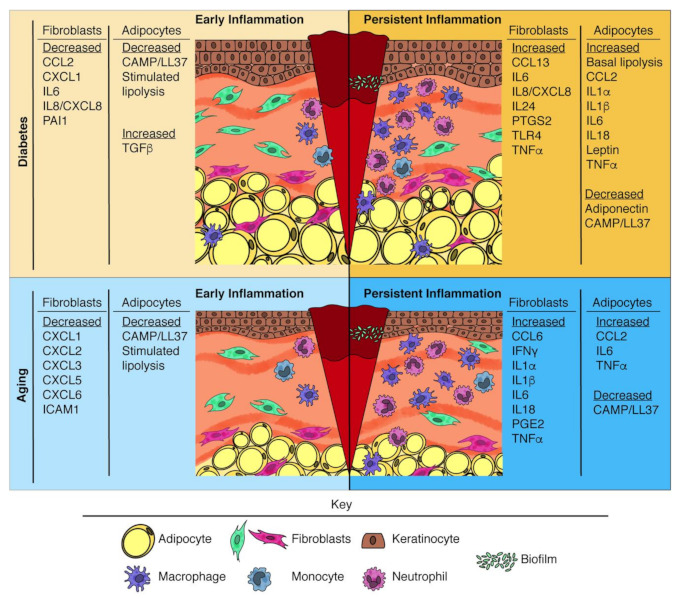
Changes in mesenchymal cell-derived immune regulators during impaired wound healing. Diagrams show representative changes to diabetic and aged skin. Diabetic skin undergoes expansion of the dermal white adipose tissue (DWAT) and a reduction in fibroblasts. Aged skin is thinner, with flatter keratinocytes, diminished DWAT, and fewer fibroblasts. Initially after injury, there is an impaired initial activation and recruitment of leukocytes to the site of injury. At later time points after injury, there is a persistence of inflammatory neutrophils and macrophages. Panels designate changes in pro- and anti-inflammatory factors from fibroblasts and adipocytes that can contribute to the altered leukocyte responses that occur with diabetes and age.

**Table 1 ijms-22-01933-t001:** Adipocyte- and fibroblast-based therapeutic targets to improve wound healing.

Inflammatory Defect during Wound Healing	Mesenchymal Cell-Based Approach to Treat Inflammatory Defect	
	Adipocytes	Fibroblasts
Impaired early myeloid cell recruitment	↑ Stimulated lipolysis ↑ Cathelicidin (CAMP/LL37)	↑ Chemokine expression (CCL2, CXCL1, CXCL2, IL8/CXCL8)
Persistent inflammation	↓ Basal lipolysis ↑ Cathelicidin (CAMP/LL37) ↑ Anti-inflammatory adipokine expression (Adiponectin) ↓ Pro-inflammatory adipokine expression (CCL2, IL1, IL6, IL18, Leptin, TNFα)	↓ Pro-inflammatory cytokine expression (IL1, IL6, IL8/CXCL8, TNFα) ↓ SASP

## Data Availability

All data generated or analyzed during this study are included in this published article.

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
