# Peer review of "Dermal Drivers of Injury-Induced Inflammation: Contribution of Adipocytes and Fibroblasts"

_ijms, 2021, doi:10.3390/ijms22041933_

Round 1

Reviewer 1 Report

The manuscript provides a very comprehensive overview of an advanced perspective regarding skin inflammation in response to injury. In particular, it is described the role of adipocytes in WAT and dermal fibroblasts in the active cross-talk with immune cells and keratinocytes and in shaping the inflammatory response and tissue repair.

The review is very comprehensive, well-structured and written. It provides a detailed overview of the role of adipocytes and fibroblasts in the inflammatory response upon injury, in the cutaneous compartment and possibly in other tissues.   The authors provide information about gene expression and protein expression in healthy conditions, in aging and in diseases such as diabetes. It is also mentioned the role of non-coding RNA and epigenetic mechanisms in regulating the inflammatory of these cells (mainly fibroblasts).

Of particular interest is the role of adipocytes of White Adipose Tissue (WAT) in producing inflammatory molecules and/or switching towards tissue repair in conditions such as diabetes or aging. This is a still relatively novel field and this manuscript will surely bring the attention of the readers on it by collecting the most relevant and advanced information and by pointing pout the potential for development of studies in this areas.

For these reasons, beside very minor spelling errors I think the manuscript is acceptable for publication as it is.

The review is well written and supported by an extensive analysis of literature data. 

Author Response

The reviewer’s considerations and kind words are very much appreciated by the authors. We have addressed spelling and grammar errors that were specifically requested by the reviewers and additional ones discovered during the review process.

Reviewer 2 Report

I have read with interest the review-paper entitled “Dermal drivers of inflammation: contribution of adipocytes and fibroblasts to injury-induced inflammation”. The article is well structured and the general idea sounds attractive. The review contains substantial degree of interest as a holistic approach of adipocyte and fibroblast functions in injury-induced inflammatory scenarios. I recommend the publication of the manuscript after the following minor revision.

MINOR POINTS:

  1. It will be highly interesting to add some information about the role of mesenchymal stem cells in the healing process (introduction). It has been described that the effects of mesenchymal stem cells are extremely important in wound healing scenarios being significant players of the switching of M1 to M2 macrophage transition among other inflammatory/immunomodulatory-related effects. Some interesting papers in this regard could be:
    1. Las Heras et. al. Chronic wounds: Current status, available strategies and emerging therapeutic solutions. Journal of Controlled Release, 328, 2020, 532-550 (https://doi.org/10.1016/j.jconrel.2020.09.039.)
    2. Motegi et al. Mesenchymal stem cells: The roles and functions in cutaneous wound healing and tumor growth. Journal of Dermatological Science, 86, 2017, 83-89. (https://doi.org/10.1016/j.jdermsci.2016.11.005.)
    3. S. Hu et al. Mesenchymal Stromal Cells and Cutaneous Wound Healing: A Comprehensive Review of the Background, Role, and Therapeutic Potential. Stem Cells International, 2018, 2018, 6901983. (doi: 10.1155/2018/6901983.)

  1. Some grammar mistakes should be corrected:
    1. In page 3 a parenthesis is opened but never closed: “express the chemokines CCL2 (monocyte chemoattractant protein 1 (MCP1), and IL8/chemokine (C-X-C motif) ligand 8 (CXCL8)”.
    2. In page 4, authors use the word correction instead of correlation: “In vivo, a positive correction exists between adipocyte size and macrophage numbers”
    3. In page 7, Il1 should be written IL1: “keratinocyte-derived Il1α induces dermal fibroblast secretion”

Author Response

We thank the reviewer for their kind words and enthusiasm for our manuscript. Below we detail the modifications made to the text that address the documented concerns.

1. We greatly appreciate the reviewer’s insight on the significance of mesenchymal stem cells in regulating inflammation during wound healing, specifically through altering macrophage polarization. We found this topic to be worthy of mentioning in multiple sections (lines 52, 433 and 459). Since our review aims to focus on tissue resident adipocytes and fibroblasts, we have elected to keep the discussion of MSCs minimal, as this topic is deserving of its own review article.

2. The reviewer’s keen eye also caught grammatical errors that have been rectified.

Reviewer 3 Report

Cooper and colleagues summarized up-to-date and very comprehensive aspects of dermal drivers of inflammation focused on adipocytes and fibroblasts. I have some comments to improve the manuscript. 

The title seems to be too broad. I would suggest revising the title to 'Dermal drivers of injury-induced inflammation: contribution of adipocytes and fibroblasts" rather than overall inflammation. 

As COVID-19 is not generally associated with impaired wound healing, the statements regarding COVID-19 (line 73-78) are not relevant in this context. I recommend removing these statements. 

line 102-104; Please supplement the examples of adipokines from subcutaneous fat rather than those from visceral fat. 

line 134: (ICAM1) IL6 -> (ICAM1), IL6

Please supplement some relevant literature regarding adipokines and interactions between fibroblasts and adipocytes upon UV irradiation and skin aging.

Please provide some therapeutic strategies and summarizing figure/table (if possible) for effective wound healing by regulating adipocytes and fibroblasts' function.

Author Response

We thank the reviewer for their support and excellent comments. Below we detail how these comments were addressed in the current version of the manuscript.

Cooper and colleagues summarized up-to-date and very comprehensive aspects of dermal drivers of inflammation focused on adipocytes and fibroblasts. I have some comments to improve the manuscript.

  1. The title seems to be too broad. I would suggest revising the title to 'Dermal drivers of injury-induced inflammation: contribution of adipocytes and fibroblasts" rather than overall inflammation.

This is an excellent suggestion and we have changed the title accordingly.

  1. As COVID-19 is not generally associated with impaired wound healing, the statements regarding COVID-19 (line 73-78) are not relevant in this context. I recommend removing these statements.

We agree that COVID-19 is not associated with impaired wound healing. We desired to bring attention to the fact that comorbidities associated with severe COVID-19 responses are associated with impaired wound healing. In order to prevent confusion or distraction from the main scope of the review, we have taken this suggestion and deleted the sentences from the review.

  1. line 102-104; Please supplement the examples of adipokines from subcutaneous fat rather than those from visceral fat. 

The authors are grateful for the reviewer’s discernment between omental and subcutaneous fat. We have added examples of adipokines that are recognized in subcutaneous adipocytes (line 98).

  1. line 134: (ICAM1) IL6 -> (ICAM1), IL6

We appreciate the reviewer’s astute eye and have fixed the grammatical error, in addition to others that were identified during editing and the review process.

  1. Please supplement some relevant literature regarding adipokines and interactions between fibroblasts and adipocytes upon UV irradiation and skin aging. (PMID: 27161953 and PMID: 28845522)

Adding context for how adipocytes and fibroblasts can interact is a great suggestion. These interactions can influence the ability of cells to be pro-inflammatory or profibrotic. Without being too speculative for how these interactions contribute to injury-induced inflammation, we have mentioned this and included the relevant literature to expand the scope of the review (Section 3.5 Communication Between Adipocytes and Fibroblasts).

  1. Please provide some therapeutic strategies and summarizing figure/table (if possible) for effective wound healing by regulating adipocytes and fibroblasts' function."

The reviewer’s request for emphasis on therapeutic strategies is of great importance to the authors and most readers of this journal. We thank the reviewer for this suggestion and have included Table 1 to summarize prospective changes to dermal adipocyte and fibroblast function that may improve healing through their contributions to wound-induced inflammation. We have attempted to keep this table concise so that it highlights the most important and practical therapeutic targets without being too redundant with what is in the text. This table is discussed in the conclusions section that is now titles “Conclusions and Future Directions”.

Reviewer 4 Report

The review is a very interesting paper on the contribution of fibroblast and adipocytes to dermal inflammation. I really enjoyed reading it. However, I have some queries and I think it needs at least a round of revision.

  • Although the paper may be considered as a narrative review, the main problem of this paper is the lack of a material and methods section describing how you selected the studies to include in this article. So in my opinion you should indicate what database you searched, what keywords you used, and a flowchart describing study exclusion and selection.
  • page 1 line 35-43 some reference for this introduction paragraph should be added; Here I suggest you some studies:doi: 10.1089/photob.2020.4908.

Thank You

Author Response

We thank the reviewer for their kind words regarding their interest in the topic of the manuscript. We agree that this is a narrative review and have mentioned this in the methods section that was added to the end of the document to describe the search terms and criteria for inclusion of cited literature.

We appreciate the suggestion to add more references and slightly broaden the scope. We have referenced reviews that describe the well accepted statements in the introduction. The idea of how UV aging impacts skin was proposed by a different reviewer and briefly discussed in its relationship adipocyte function, including the potential for adipocytes to influence fibroblast function. While this is different from the proposed study, we believe this is more directly relevant to how adipocytes and fibroblasts are able to influence the early inflammatory events after injury.

Round 2

Reviewer 4 Report

The authors responded to all queries. The article is ready to be published.